# Expanding the Clinical Phenotype of 19q Interstitial Deletions: A New Case with 19q13.32-q13.33 Deletion and Short Review of the Literature

**DOI:** 10.3390/genes13020212

**Published:** 2022-01-24

**Authors:** Elena-Silvia Shelby, Michael Morris, Liliana Pădure, Andrada Mirea, Relu Cocoș, Alexandru Cărămizaru, Simona Șerban-Sosoi, Andrei Pîrvu, Ioana Streață

**Affiliations:** 1National University Center for Children’s Neurorehabilitation “Dr. Nicolae Robănescu”, 44 Dumitru Mincă Street, District 4, 041408 Bucharest, Romania; silviajdx@yahoo.com (E.-S.S.); lilianapadure@gmail.com (L.P.); 2Department of Genetics, SYNLAB Switzerland, Chemin d’Entre-Bois 21, 1018 Lausanne, Switzerland; michael.morris@synlab.com; 3Faculty of Midwifery and Nursing, University of Medicine and Pharmacy “Carol Davila”, 8 Eroilor Sanitari Boulevard, 050474 Bucharest, Romania; 4Chair of Medical Genetics, University of Medicine and Pharmacy “Carol Davila”, 37 Dionisie Lupu Street, 020021 Bucharest, Romania; relu_cocos@yahoo.com; 5Human Genomics Laboratory, Craiova University of Medicine and Pharmacy, 2-4 Petru Rareș Street, 200349 Craiova, Romania; alexandru.crgm@gmail.com (A.C.); simona.sosoi@umfcd.ro (S.Ș.-S.); andrei.crgm@gmail.com (A.P.); ioana.streata@yahoo.com (I.S.); 6Regional Centre of Medical Genetics Dolj, Clinical Emergency County Hospital, Craiova, 1 Tabaci Street, 200642 Craiova, Romania

**Keywords:** 19q13 microdeletion syndrome, array-CGH, contiguous gene syndrome

## Abstract

19q13 microdeletion syndrome is a very rare genetic disease characterized by pre- and postnatal growth retardation, intellectual disability, expressive language impairment, ectodermal dysplasia, and slender habitus. Since the description of the first case in 1998, less than 30 cases have been reported worldwide. This article aims to review the knowledge gathered so far on this subject and to present the case of a 10-year-old girl admitted to the National University Center for Children Neurorehabilitation “Dr. Nicolae Robanescu” in November of 2018 who presented a slender habitus, growth retardation, facial dysmorphism, skeletal abnormalities, and ectodermal dysplasia. Array-CGH analysis revealed a 1.53 Mb deletion in the 19q13.32-q13.33 region. MLPA for the FKRP gene revealed that the microdeletion was de novo. The patient’s phenotype overlapped with the clinical features of 19q13 microdeletion syndrome. To our knowledge, this is the first case of 19q13 microdeletion syndrome to ever be reported in Romania. We believe our case presents additional features that have never been previously reported in this syndrome, namely, dilatation of the third ventricle and subependymal cyst, left iris coloboma, and tracheomalacia. Moreover, unlike the other 19q13 microdeletion cases that presented with dystonia, our patient also presented dystonia but, interestingly, without having haploinsufficiency of the KMT2B gene.

## 1. Introduction

19q13 microdeletion syndrome is a very rare genetic disease characterized by pre- and postnatal growth retardation, intellectual deficit, expressive language impairment, slender habitus, and ectodermal dysplasia [1]. The known number of 19q13 microdeletion cases is very small, possibly due to the fact that small rearrangements are hard to detect in this region, which lacks a distinct banding pattern [2,3]. With the advent of array-CGH platforms for genome screening [4], identifying and characterizing contiguous gene syndromes caused by submicroscopic deletions in this region is becoming increasingly easier [4].

## 2. 19q13 Microdeletion Syndrome—A Review of the Literature

Related to the 19q13.2 locus, it is well known that haploinsufficiency of the RPS19 gene [5] is involved in the occurrence of Blackfan–Diamond syndrome, a disorder characterized by bone marrow failure with subsequent severe anemia [6].

The first report of a 19q13 deletion was in 1998 in a patient with IUGR and decreased fetal activity [7]. Karyotype from amniotic fluid (GTG banding) suggested a deletion in the 19q13 region: del(19)(q13.1q13.3) [5]. Subsequent microsatellite analysis revealed a 15 Mb deletion involving the distal portion of q12 and the proximal portion of q13.1 (19q12.1-q13.1 deletion). The patient, a girl, was severely affected (a single umbilical artery, scarce subcutaneous fat, poor feeding requiring a gastrostomy tube, congenital deafness, hypotonia, aplasia cutis of the posterior fontanel, clinodactyly of the fifth finger, overlapping toes, cardiomegaly, congenital deafness, and severe motor and cognitive delay, as well as facial dysmorphism, with a long face; micrognathia; posteriorly rotated, low-set ears; a high palate; hypertelorism; and a broad nasal root) [7,8].

Two years later, Tentler et al. [9] hypothesized that a deletion of 3.2 Mb in the 19q13.2 region could possibly represent a contiguous gene syndrome associated with Blackfan–Diamond anemia, skeletal malformations, and mental retardation.

In 2009 [7], three unrelated patients with overlapping 19q13.11 deletions were described. Patients shared common features: a slender habitus, pre- and postnatal growth retardation, microcephaly, feeding difficulties, hypospadias, and various signs of ectodermal dysplasia, such as occipital aplasia cutis, sparse and thin hair, sparse eyebrows, and dysplastic nails. They also presented mild facial dysmorphism, with a long face, retrognathia, a high forehead, thin lips, large ears, and a V-shaped nose with hypoplastic alae, as well as widely spaced nipples, a single umbilical artery, long and tapered fingers, and a single median incisor [7,9]. A critical region of 2.87 Mb was established. As breakpoints were identified in these patients, it was concluded that the mechanism of non-allelic homologous recombination was not responsible for the deletion [3]. This report reinforced the fact that this microdeletion could represent a clinically recognizable genetic syndrome [3].

In the same year, Schuurs-Hoeijmakers et al. [10] reported a fifth patient with an interstitial deletion overlapping the 19q13.11 region. The patient was a boy from a twin dizygotic pregnancy obtained from non-consanguineous parents, born at 37 weeks and small for gestational age (birth weight 1620 g, −3.5 SD), with IUGR. His sister was normal for gestational age, weighing 2290 g (25th percentile), and had no constitutional abnormalities. At birth, the patient presented hypospadias, occipital aplasia cutis, dysplastic nails, long fingers with slender thumbs, abnormally positioned feet, a sacral dimple, and a third nipple, which was inherited from the father [8,10]. The patient also had facial dysmorphism, with deep-set eyes, a large mouth with thin lips, a wide nasal bridge, retrognathia, abnormal position of the teeth, and broad gums [8]. After birth, he presented feeding difficulties, failure to thrive, and delay in obtaining motor and cognitive acquisitions: He could sit independently at two years of age and could walk without help at three years and 10 months, being able to walk only for short distances, with gait imbalances [10]. At the age of four years and 10 months he had not acquired speech, communicating only with pictograms [10].

In 2010, Adalat et al. described a boy with prenatal and postnatal growth retardation with microcephaly, fifth-finger clinodactyly, bilateral pes cavus, a narrow thorax with an extra nipple, and dysmorphic features such as a narrow face, a narrow nasal bridge, blepharophimosis, thin lips, a short philtrum, large and posteriorly rotated ears, occipital aplasia cutis, sparse hair, small nails, a lack of eyelashes on the lower lids, and crowded teeth. He had wheezing and recurrent infections in the first year of life. He required surgery for pyloric stenosis at one month of life and for intestinal intussusception at one year. Postnatal imaging showed that he had short, echogenic kidneys, with the left kidney ensuring 87% of the function [11].

By 2012, 19q13.11 deletion syndrome was already recognized as a new genetic syndrome characterized by pre- and postnatal growth retardation, genital malformations (hypospadias) in males, mental retardation, microcephaly, developmental delay, and speech disturbance in the context of normal family history and non-consanguineous parents [4]. That year, the small number of cases already known was enlarged following an article published by Gana et al. [4] that presented two patients with 19q13.11 deletion with the already recognizable pattern of malformations described above. Besides the two cases presented, which added up to the five previously published cases and the two cases reported in the DECIPHER database (Database of Chromosomal Imbalance and Phenotype in Humans Using the Ensemble Resources), the novelty of the article consisted of the fact that it further refined the minimal overlapping region (MOR) for this 19q13.11 deletion to 324 kb [4], comprising six genes: ZNF302, ZNF181, ZNF599, and ZNF30—four zinc-finger (ZNF) genes of the KRAB-ZNF (Kruppel-associated box—ZNF) subfamily with the role of ubiquitous transcription repressors, as well as two non-coding RNA genes (ndRNA), namely, LOC400685 and LINC00904, whose role is still unknown [1]. It was presumed that the haploinsufficiency of these six genes is responsible for this contiguous gene syndrome [4,12].

The study also showed that larger deletions in this area, which comprise the WTIP gene encoding the Wilms tumor interacting protein, are responsible, through this protein’s interaction with WT1, for hypospadias in males [4]. Based on finding this mutation in a patient with a 46,XX karyotype, bilateral cryptorchidism, and WTIP gene mutation, it was hypothesized that mutations in this gene can also be responsible for sex reversal [4].

Among the two cases presented in this article, besides the already known clinical features (pre- and postnatal growth retardation, skeletal abnormalities such as syndactyly and fifth-finger clinodactyly, hypospadias, microcephaly, severe mental retardation, aplasia cutis), one of them presented with a ventricular septal defect detected before birth, fetal distress that required caesarean section at 34 weeks of pregnancy, minor facial dysmorphism (long and narrow face; thick eyebrows that were medially sparse; thin lips; retrognathia; large, low-set ears; hypertelorism), and growth hormone deficiency [4,8]. The patient had aplasia cutis in the middle of the scalp [10]. He also presented dystonia, which started in the lower limbs at the age of four and progressed to generalized dystonia complicated by tremors, which had an unsatisfactory response to oral treatment with pimozide (the patient had a long QT episode), botulinum toxin, and intrathecal baclofen [4]. The patient also showed dystonic tetraplegia with subsequent joint dislocations and muscle contractures [9]. A brain MRI revealed enlarged lateral ventricles. Prior to the CGH array, karyotype, subtelomeric FISH analysis, FISH for DiGeorge syndrome and DYT1 mutation analysis had normal results [4].

The second patient presented by this study, a girl, had the same clinical features—prenatal and postnatal growth retardation, failure to thrive, global delayed development, and microcephaly—as the previously described patients [4]. Furthermore, the girl had abnormal tongue movement and a hoarse voice. She also had gait abnormalities, mild hypotonia, astigmatism, and facial dysmorphism, with a narrow face, thick eyebrows that were medially sparse, up-slanting palpebral fissures, thin lips, short philtrum, micrognathia, hypodontia, and large ears. She also presented bilateral brachydactyly of the fifth fingers [4] and dental abnormalities [8].

Furthermore, because no genes were identified to be related to dystonia in these patients, the authors of this study hypothesized that, here, dystonia appears according to the two-hit model, with the synergic effect of several genes as a predisposing factor and various genetic, epigenetic, or environmental effects as the decisive factors in producing this symptom [4].

Moreover, as the 19q13.11 deleted region includes the SCN1B gene, which encodes a subunit of a sodium channel, and it is known that heterozygous mutations in this gene are responsible for heart defects, generalized epilepsy with febrile seizures, and Brugada syndrome type 5, the authors presumed that haploinsufficiency of this gene might require special care in the administration of neuroleptics such as pimozide [4].

In the same year, Forzano et al. [2] presented a patient with a 1.37 Mb deletion in the 19q13 region who had all the characteristics of this syndrome, as well as multiple pituitary hormone deficiency. The patient had intellectual disability; facial abnormalities such as asymmetry, orofacial grooves, ptosis, and eye movement abnormalities; scoliosis; heart defects; and chronic constipation [2].

Two years later, in 2014, Chowdhury et al. [13] presented five patients with deletions in the 19q12-q13.1 region. Among them, two had deletions overlapping the SLO (smallest region of overlap) and three had deletions proximal to the SLO. An unrelated boy and girl with deletions overlapping the SLO had a 19q13 microdeletion phenotype, namely, prenatal and postnatal growth retardation; feeding difficulties; overall delayed development, including delayed speech; finger and toe abnormalities; and facial dysmorphism, with short palpebral features, arched eyebrows, a short nose with a broad bridge and upturned nares, a smooth philtrum, and misfolded ears. The boy had a single umbilical artery, scalp aplasia cutis, cryptorchidism, hypospadias, an inguinal hernia, febrile seizures, and an atrial septal defect [13]. The girl had caries and dental crowding [13].

In another article released towards the end of 2014, Castillo et al. [14] presented three unrelated patients who had in common dysmorphic features and growth retardation, all having an interstitial microdeletion of various sizes of the 19q13.32 region [14]. The first patient, with a 1.3 Mb deletion, had facial dysmorphism, with facial asymmetry with submucous palatoschizis, micrognathia, slanted palpebral fissures, bilateral ptosis, oculomotor palsies, corpus callosum aplasia, kyphoscoliosis, increased inter-nipple distance, a right-side-oriented aortic arch, a bilateral inguinal hernia, a double nail at the second toe of the left leg, partial syndactyly of the second and third toes, and bowel atony [14].

This study [14] concluded that the common abnormalities presented as part of this syndrome (facial asymmetry with orofacial grooves, ptosis, oculomotor palsies, micrognathia, aortic abnormalities, kyphoscoliosis, and bowel atony) are most likely the result of haploinsufficiency of approximately 20–23 genes, such as NAPA, NPAS1, ARHGAP35, DHX34, SLC8A2, MEIS3, and ZNF541. These genes are expressed in various tissues and organs, such as the brain parenchyma, musculoskeletal system, heart, and gastrointestinal tract [14].

In the same year, December 2014, Venegas-Vega et al. [12] highlighted the small number of cases of 19q13 microdeletion syndrome known at that point—11 in the entire world—and presented the first case of 19q11.13 microdeletion syndrome caused by a chromosomal rearrangement [12]. The patient, a boy of non-consanguineous parents, had two siblings. The mother had a miscarriage prior to his birth. Similar to the other patients, the boy had low birth weight (>3rd percentile) and length (10th–25th percentile), feeding difficulties, a slender habitus, developmental delay, hypospadias, a shawl scrotum, clubfeet in a varus position, hip dislocation, aplasia cutis in the scalp region, recurrent upper airway infections, and slightly dysmorphic facies with long face, high forehead, hypoplastic alae nasi, sparse eyebrows and eyelashes, and low-set ears. At the age of five, the patient had one febrile seizure and underwent surgical interventions for bilateral cryptorchidism and bilateral inguinal hernia [12]. At the age of six years and seven months, he had severe growth retardation (height and weight under the 3rd percentile). Hormonal tests and a pelvic ultrasonogram, EEG, echocardiogram, and audiometry had normal results [12]. Karyotype banding (GTG) showed normal and derivative chromosomes 2 and 19, with the insertion of the segment from 19q13.12 to 19q13.43 (with concomitant deletion in this area) in 2p25.3. The rearrangement was further confirmed by FISH testing [12].

The article also highlights that, in 10 out of the 11 patients, besides the 324 kb critical region, the deletion also comprised three other genes, namely, UBA2, WTIP, and SCGB2B2 [1]. SCGB2B2 is a member of the secretoglobulin family of proteins, which are known to have immunomodulatory roles. WITP was proposed as a candidate gene for hypospadias. The authors presumed that haploinsufficiency of UBA2 contributes to the genital abnormalities [12]. This is the first study that states that UBA2, WTIP, and SCGB2B2 might be responsible for phenotypic features of this syndrome [12].

In 2017, Meyer et al. [15] presented 10 unrelated patients with 19q13.11-q13.12 microdeletions that appeared de novo and varied in size, from 14 to 100 genes, with the smallest region of overlap between these microdeletions including the ZBTB32 and KMT2B genes, the latter being, in the authors’ opinion, responsible for the dystonia symptoms. In all of the patients, the deletion was spontaneous and all of them presented with progressive dystonia with onset between 2.5 and seven years of life, affecting the lower limbs and (in all patients except for one) upper limbs, the cervical area, and the face, and with the patients having dysarthria, dysphonia, and chewing and swallowing difficulties [15]. The patients had abnormal signals in the globus pallida. One of the patients had pronounced craniofacial dysmorphism, namely, microcephaly, epicantus, blepharophimosis, low-set and posteriorly rotated ears, narrow nasal bridge, dental crowding, micrognathia, and occipital aplasia cutis. Two patients had a bulbous nose and six had an elongated face. The other patients had rare dysmorphic features, one in each patient: short stature, seizures characterized by absence, ectodermal dysplasia, a hyperechogenic kidney, absent testis, and retinal dystrophy [15]. Eight out of the 10 patients had various degrees of mental retardation [15].

Located in the same region, in the 19q13.12 locus is WDR62–WD-REPEAT CONTAINING PROTEIN 62, a microtubule-associated protein involved in centrosome biogenesis and normal cell division [16,17]. As shown by Bilguvar et. al, as well as other authors [17,18,19,20,21], WDR62 has a very important role in neurogenesis and brain development; biallelic pathogenic mutations in this gene are responsible for primary microcephaly-2 (MCPH2) with or without cortical malformations.

At the present time, according to OMIM, Orphanet, and GARD (Genetic and Rare Diseases Information Center), chromosome 19q13.11 deletion syndrome (OMIM #613026) is characterized by global growth failure; IUGR; feeding difficulties; cachexia; developmental delay; mental retardation; speech difficulties; ectodermal dysplasia with dry skin, sparse lateral eyebrows, sparse eyelashes, thin and sparse hair, or hypodontia; facial dysmorphism with a long face, a high forehead, retrognathia, a thin vermilion border, underdeveloped nasal alae, cataract, short palpebral fissures, blepharophimosis, a wide mouth, a single median maxillary incisor, a long palpebral fissure, microcornea, macrotia, a short nose, a wide nasal bridge, or anteverted nares; a single umbilical artery; limb abnormalities, including fifth-finger clinodactyly, overlapping toes, or cutaneous syndactyly; abnormal cardiac septum morphology; widely spaced nipples; and, in males, genitourinary abnormalities like hypospadias, cryptorchidy, or bifid scrotum [1,8,22]. Recurrent infections, febrile seizures, inguinal hernias and hearing impairment are also described [22].

Orphanet further describes the facial dysmorphism as minor, with a long and narrow face, high forehead, V-shaped nasal tip, retrognathia, prominent columella, hypoplastic alae nasi, and large ears [12]. All cases described have been sporadic, with no family history of the disease [22].

Besides the previously described 19q13.11 distal deletion syndrome (OMIM #613026), OMIM also describes a proximal 19q13.11 deletion syndrome (OMIM #617219) characterized by delayed development, mental retardation with speech difficulties, autistic features, and, in some cases, renal abnormalities [1].

## 3. First Case of 19q13.32-q13.33 Deletion to Be Reported in Romania

Here we present the case of a 10-year-old girl admitted at National University Center for Children Neurorehabilitation “Dr. Nicolae Robanescu” in November 2018 for evaluation and medical rehabilitation.

Family history was normal, and the parents were not consanguineous. The patient came from a GIPI mother, with term delivery, 17 h-long labor, and vaginal birth. The patient was appropriate for gestational age, with no prenatal growth retardation (birth weight was 3300 g—50th percentile; birth length was 50 cm—50th percentile).

According to the discharge documents from the maternity ward, the patient had severe perinatal hypoxia due to meconium aspiration syndrome, presenting transient seizures that have not been repeated since discharge from the maternity ward, discrete laryngeal stridor, physiologic jaundice, decreased skin turgor, decreased general tonus and reactivity, atrial and ventricular septal defect, left iris coloboma, congenital tracheomalacia, subependymal cerebral cyst and moderate enlargement of the third ventricle (as seen on a transfontanellar ultrasound), facial dysmorphism, and left pelvicalyceal duplication. Hypoxia was remitted following oxygen administration. The electrolyte imbalance was corrected and antibiotic therapy using intravenous ampicillin and gentamicin was administered for five days.

Prior to admission at National University Center for Children Neurorehabilitation “Dr. Nicolae Robanescu,” the patient had been investigated in other hospitals, where she had received ENT, cardiologic, ophthalmologic, and genetic consults, as well as transfontanellar and abdominal ultrasounds. These investigations are presented in the following paragraphs.

The ENT consult revealed tracheomalacia in the middle third of the trachea, especially in the left half of the lumen.

The pediatric cardiology consult revealed a slight dilatation of the pulmonary trunk with retrograde flux, cardiac valves with hyperechogenic commissures, an interatrial septum with foramen oval, a small ventricular sept muscular defect, small subaortic communication represented by a perimembranous ventricular septal defect, stage II–III tricuspid regurgitation caused by tricuspid valve dysplasia, small arterial canal persistence with no hemodynamic significance, and bilateral ventricular hypertrophy.

The transfontanellar ultrasound revealed a moderate dilatation of the third ventricle and the presence of a left subependymal cyst. The corpus callosum, median line structures, choroid plexus, and posterior fossa had normal conformation.

An abdominal ultrasound revealed the presence of left pelvicalyceal duplication. The liver, spleen, and pancreas had normal dimensions; the gallbladder was slightly plied; the right kidney was normally conformed; and the left kidney presented pelvicalyceal duplication, with a slight dilatation of the inferior renal pelvis.

The ophthalmologic exam revealed a normal funduscopic exam; a bilaterally normal anterior chamber of the right eye; and iris coloboma in the left eye.

The patient’s first genetic evaluation was done in 2008. The patient had a normal 46, XX karyotype (genetic evaluation and karyotype were done at the Medical Genetics Department of Alessandru Rusescu National Institute for Mother and Child’s Health). In September 2018, in the same hospital she was tested for Wolf–Hirschhorn syndrome, using FISH, with normal results. Consequently, the karyotype was repeated and the result came back normal.

Following evaluation in our hospital, we noted that the patient presented total kyphosis, bilateral talus valgus, generalized hypotonia, ligamentous hyperlaxity, predominantly distal motor deficit, L5 spina bifida, and severe intellectual disability, with an IQ under 30.

The patient presented delay in obtaining motor and mental acquisitions. She could sustain her head at about three months, sit without help at seven months, walk with bilateral support at about six years, and at the time of the consult, ambulation was possible only with unilateral support. The patient was severely intellectually disabled and had speech retardation, as well as a very particular way of pronouncing the words—precipitously and almost unintelligibly. She also presented hand stereotypies (hand clapping, dystonic positions of the fingers). The patient presented agitation, restlessness and hyperactive behavior, and, according to the mother, oftentimes she displayed self-aggressive behavior and low tolerance at frustration.

The gait was unstable; the patient walked with a wide stance and stiff legs, with the tips of her feet pointed towards the exterior. The patient could descend stairs only by “falling’’ on one leg or the other, holding the legs stiff.

According to the mother, the patient’s hearing was normal.

At the clinical exam, the patient had a weight of 20 kg (below the 1st percentile) and a height of 128 cm (3rd percentile). The head circumference was 49.5 cm (two standard deviations lower than normal). The patient thus presented postnatal growth retardation and a very slender body type, as well as microcephaly.

Facial dysmorphism was seemingly non-specific, although she had a very prominent nose (almost “beak-shaped’’); down-slanting palpebral fissures; bilateral palpebral ptosis; low-set and misshapen ears with abnormal pavilions; micrognathia; retrognathia; a short philtrum; left-eye coloboma; a high, arched palate; a wide nasal bridge; columella under the nasal alae; an everted lower lip; sparse, brittle hair; dental dystrophy with abnormally inserted canine teeth; bilateral dystrophy of the hallux nails; thick eyebrows that were sparse towards the midline; and a high forehead with a high hairline (Figure 1 and Figure 2). She also presented dorsolumbar gibbosity, pectus excavatum, total kyphosis, bilateral clinodactyly of the fifth fingers of the hands, bilateral talus valgus (right more pronounced than left), hypotonia, skin hyperlaxity, hyperextensible joints, and widely spaced nipples (Figure 3).

In November 2018, at the National University Center for Children Neurorehabilitation “Dr. Nicolae Robanescu,” Bucharest, the parents signed the informed consents and peripheral venous blood was obtained from the patient in order to perform an aCGH (array comparative genomic hybridization) analysis at the Clinical Regional Center of Medical Genetics Dolj (CRGM Dolj) in Craiova. Leukocytes from peripheral venous blood collected on EDTA were used to obtain genomic DNA, according to the protocol of the Wizard^®^ Genomic DNA Purification Kit (Promega, Madison, WI, USA). An Eppendorf Biophotometer was used to assess DNA purity and concentration. Following extraction, the DNA sample was further processed in order to detect copy number variations (CNVs) according to the Agilent protocol (Version 7.4 August 2015) using reference male and female genomic DNA (Agilent Technologies, Human Reference DNA, male and female). High-resolution microarray analysis was performed with the Agilent SurePrint G3 ISCA v2 CGH + SNP array 4 × 180 k microarrays. All the protocol steps (restriction digestion, fluorescent labeling, purification of both reference and patient gDNA, hybridization, microarray wash, and microarray scanning and analysis) were performed according to the manufacturer’s protocol (Agilent protocol—Version 7.4 August 2015). The array images were acquired using the NimbleGen MS 200 Microarray Scanner (Roche). Agilent Feature extraction software was used for data extraction. Copy number data were analyzed with Agilent Cytogenomics 5.0 software (Agilent Technologies Inc., Santa Clara, CA, USA). The current 2020 ACMG aCGH classification was used.

The genetic analysis was done free of charge through the National Health Program for Rare Diseases. A 1.53 Mb deletion in the 19q13.32-q13.33 region was found, encompassing 45 genes, among which were 27 OMIM genes, namely, PPP5C, CCDC8, CALM3, PTGIR, DACT3, PRKD2, STRN4, FKRP, SLC1A5, AP2A1, ARHGAP35, NPAS1, SAE1, BBC3, C5AR1, C5AR2, DHX34, SLC8A2, KPTN, NAPA, BICRA, EHD2, NOP53, SELENOW, TPRX1, CRX, and SULT2A1.

To confirm the microdeletion using a second, independent technique, MLPA analysis was conducted using commercially available probes for the FKRP gene, which is located within the 19q13.32-q13.33 region. The confirmation was performed on DNA isolated from EDTA blood from both the parents and the patient. We performed MLPA analysis using the Salsa MLPA Probemix P116-B2 SGC kit according to the manufacturer’s instructions (MRC Holland, Amsterdam, the Netherlands). Confirmation of the presence of the FKRP heterozygous gene microdeletion and the de novo status of the mutation was acquired by analyzing the data using Softgenetics Genemarker v.2.7.4 (Appendix A).

## 4. Discussion

The number of cases of 19q13 microdeletion syndrome reported so far is very small (to our knowledge, less than 30 such cases have been reported since the first description of this syndrome in 1998, among which, according to an article by Abe et al. from 2018 [23], 14 cases were reported with 19q13.11 deletion and an additional four such cases were annotated in DECIPHER—the Database of Chromosome Imbalance and Phenotype in Humans using Ensembl Resources).

Due to the small number of cases reported so far, the information about the clinical traits of patients with 19q13 microdeletion syndrome is scarce, and, many times, confusing, as each of the cases presented so far has had a different size of the deletion, and thus, a different—albeit partially overlapping with the other cases—phenotype.

Here, we presented another case of 19q13 microdeletion syndrome, a 10-year old girl with non-consanguineous parents and normal family history with a 1.53 Mb deletion in the 19q13.32-q13.33 region. The patient’s phenotype overlapped the phenotype of 19q13 microdeletion syndrome, as our patient presented postnatal growth retardation, a slender habitus, intellectual disability, absent speech, heart malformations (slight dilatation of the pulmonary trunk with retrograde flux, cardiac valves with hyperechogenic commissures, an interatrial septum with foramen oval, a small ventricular sept muscular defect, small subaortic communication through a perimembranous ventricular septal defect, stage II–III tricuspid regurgitation through tricuspid valve dysplasia, small arterial canal persistence with no hemodynamic significance, and bilateral ventricular hypetrophy) and kidney malformations (left pelvicalyceal duplication); kyphosis; widely spaced nipples; microcephaly; facial dysmorphism with down-slanting palpebral fissures; a long face; bilateral ptosis; a prominent nose with columella well below the alae nasi; thin lips with an everted lower lip; low-set, posteriorly rotated ears; a short philtrum; a high, arched palate; and signs of ectodermal dysplasia, namely, malpositioned canine teeth, sparse hair, thick eyebrows that were sparse towards the nose, a high forehead with a high hair line, bilateral nail dystrophy, bilateral clinodactyly of the fifth finger, and kyphosis. Moreover, this patient also presented brain malformations, left iris coloboma, and tracheomalacia, features that have never been reported in this syndrome. MLPA for the FKRP gene revealed that the microdeletion was de novo; the recurrence risk is presumed to be low although germinal mosaicism cannot be formally excluded. Although all cases of 19q13 microdeletion syndrome were de novo, the parents were advised to continue genetic testing to establish the segregation pattern and assess the recurrence risk. The parents refused testing, as they do not wish for other pregnancies.

Moreover, the patient was adopting dystonic positions of her fingers. It is already well known that dystonia is a common feature in this syndrome, and is produced, according to some authors [15], by the haploinsufficiency of the KMT2B gene. Interestingly, this patient did not have haploinsufficiency of the KMT2B gene.

Although most of the cases that led to the description of the phenotype of patients with 19q13 microdeletion syndrome consist of 19q13.11 deletions, there have been, to our knowledge, five cases reported so far in the literature of 19q13.32 microdeletions. Thus, in 2014, Castillo et al. [14] mentioned a previously described case of a de novo 732 kb microdeletion in the 19q13.32 region, a patient with intellectual disability, cardiac defects, scoliosis, facial asymmetry, oculomotor abnormalities, ptosis, and orofacial clefts. Castillo et al. reported three more unrelated patients, all of them having in common distinctive dysmorphic features and intellectual disability. Among them, the patient with the largest deletion, comprising a region of 1.3 Mb, presented hypotonia, kyphoscoliosis, facial dysmorphism with facial asymmetry, down-slanting palpebral features, micrognathia, a submucosal cleft palate, and other features, such as widely spaced nipples, a double nail on the second toe of the left foot, partial syndactyly of the second and third toes, a right-side aortic arch, hypospadias, bilateral inguinal hernias, and colonic atony. The authors concluded that the above-mentioned features are common features of 19q13.32 microdeletion syndrome, and that these features result from the deletion of 20–23 genes located in this region, among which are NPAS1, NAPA, ARGHGAP35, DHX34, SLC8A2, ZNF541, and MEIS3, genes that are expressed in the brain parenchyma, glial cells, musculoskeletal system, heart, and gastrointestinal system [14]. In 2017, Travan et al. [24] described a fifth patient with a 327 kb deletion in this region, consisting of eight genes: ARGHAP35, NPAS1, TMEM160, ZC3H4, SAE1, BBC3, MIR3190, and MIR3191. The patient had hypotonia, facial dysmorphism, micrognathia, developmental delay, kyphoscoliosis, and a buried penis [24].

After corroborating the available information in the literature, we can conclude that 19q13 microdeletion syndrome is very complex and, being a contiguous gene syndrome, each case is different even though many features overlap in all patients. We could divide this syndrome into the 19q13.11 microdeletion subgroup, with 14 patients reported so far, according to Abe et al. (2018), and the 19q13.32 microdeletion subgroup [23], with five patients reported so far, according to Castillo and Travan [14,24]. Interestingly, the features overlap, with the exception of cleft palate, right-side aortic arch, and colonic atony, which seem to be related to the 19q13.32 region [14,24]. It is worth mentioning that our patient did not present submucosal cleft palate but had a very high palate. Moreover, febrile seizure is mentioned as being part of this syndrome, albeit rare [8], and our patient presented transient seizures in the neonatal period, while she was receiving intravenous antibiotic treatment. Unlike the patients with 19q13.32 microdeletion syndrome reported by Castillo, our patient did not have syndactyly, a right-side aortic arch, colonic atony, oculomotor abnormalities, cleft palate, or inguinal hernia. Interestingly, her features overlapped the “classical” features of 19q13.11 microdeletion syndrome more. Moreover, our patient presented three features that have, to our knowledge, never been reported so far as being connected to 19q13 microdeletion syndrome, namely, tracheomalacia, iris coloboma, and brain abnormalities consisting of third ventricle enlargement and a subependymal cyst, features that could possibly be specific to the 19q13.32 microdeletion, in which the only brain abnormality mentioned is aplasia of the corpus callosum [14,24].

Related to the patient’s brain abnormalities (moderate dilatation of the third ventricle and the presence of a left subependymal cyst), an article from 2004 by Chae et al. [25] states that homozygous hyh mice (mice with a pathogenic missense mutation in the N-ethylmaleimide sensitive factor attachment protein α, also known as NAPA or SNAP-α) [26] presented a significant cystic dilation of the ventricles at birth as well as hopping gait. These mice also showed a significantly small cortex at birth and suffered progressive enlargement of the ventricular system, leading to their death [27]. Interestingly, our patient had haploinsufficiency of the NAPA gene. Nevertheless, these features could be caused by the haploinsufficiency in our patient of other genes that are present mainly in the brain, such as STRN4 [28] or NPAS1 [29].

Referring to the patient’s iris coloboma, it is known that homozygous pathogenic mutations in the FKRP gene, which was haploinsufficient in our patient, are responsible for three different forms of muscular dystrophy-dystroglycanopathies (MDDG), among which is MDDGA5, or Walker–Warburg syndrome [30,31], which can present, among other features, eye abnormalities such as iris coloboma [31].

Regarding the patient’s tracheomalacia, although some genes responsible for this trait are present in the region, such as ERF or LTBP4 on chromosome 19q13.2 [32,33], none of the genes that were haploinsufficient in our patient seemed to be responsible for this feature. It is possible this trait is the result of more subtle interactions in housekeeping genes.

## 5. Conclusions

19q13 microdeletion syndrome is a very rare genetic disease, with less than 30 cases reported since the description of the first case in 1998. To our knowledge, the case presented above is the first to ever be reported in Romania. At the same time, our patient presented additional features, namely, brain malformations consisting of ventricular enlargement and a subependymal cyst, left iris coloboma, and tracheomalacia. Interestingly, the patient was dystonic without presenting haploinsufficiency of the KMT2D gene. We believe our article is useful in furthering the knowledge about this syndrome, especially about the phenotype of 19q13.32 deletions, as well as in broadening the phenotype of this very rare disease.

## Figures and Tables

**Figure 1 genes-13-00212-f001:**
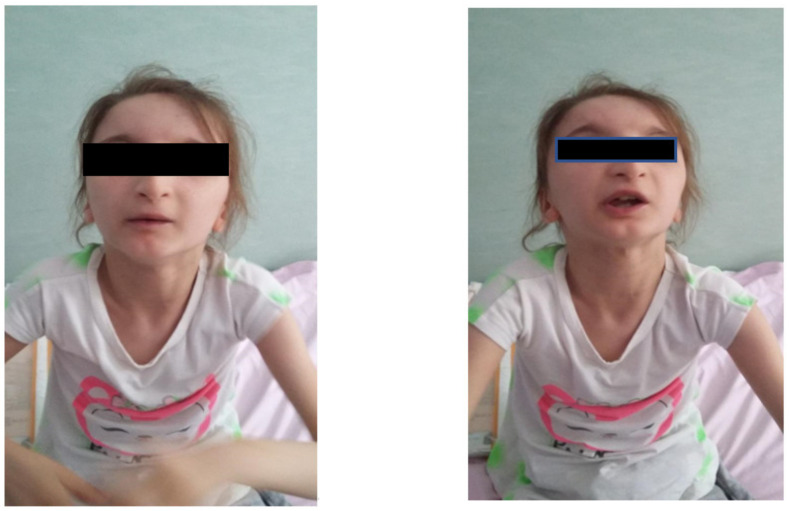
The patient had down-slanting palpebral fissures; thin lips with an everted lower lip; low-set, posteriorly rotated ears; a short philtrum; a prominent nose with columella under the alae nasi; and several signs of ectodermal dysplasia: thick eyebrows that were sparse in the middle portion, brittle hair, a high forehead with a high hairline, and malposition of the canine teeth.

**Figure 2 genes-13-00212-f002:**
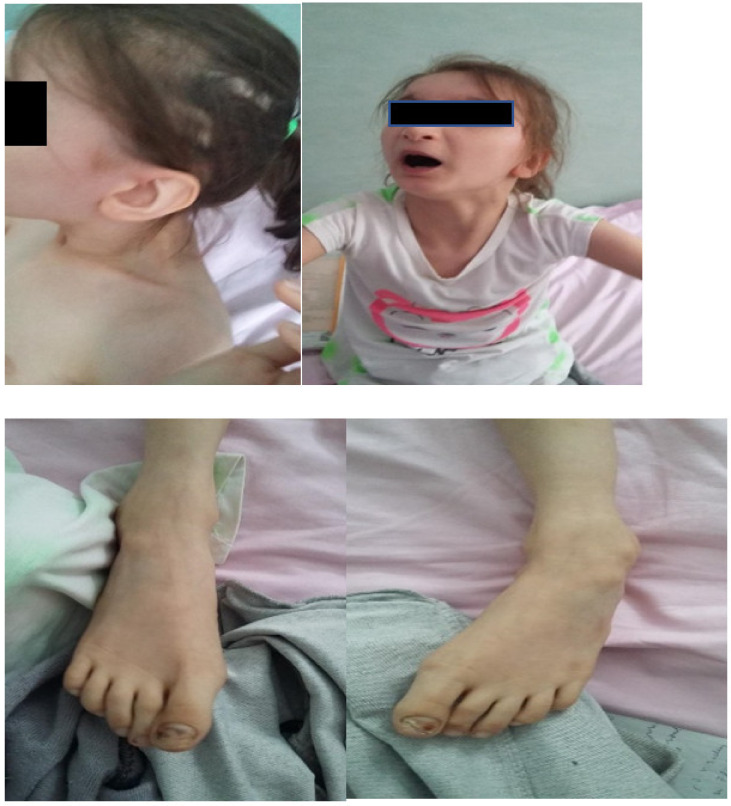
Ectodermal dysplasia, represented in this patient by sparse hair (**upper left** quadrant), thick eyebrows that are sparse in the middle (upper right quadrant), malpositioned teeth, and nail dystrophy (**lower left** and **right** quadrants).

**Figure 3 genes-13-00212-f003:**
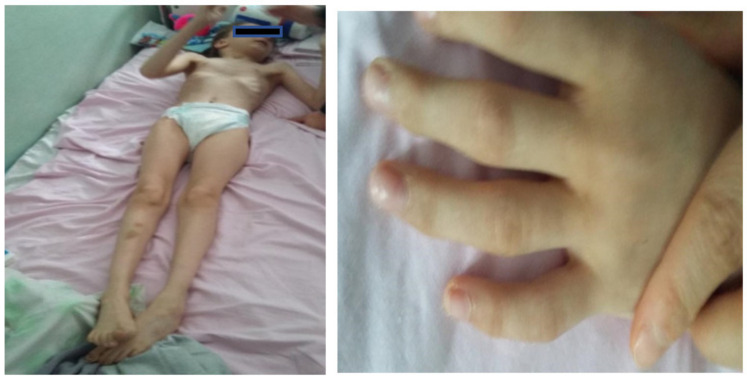
Patient had a slender habitus and bilateral clinodactyly of the fifth finger (upper limbs).

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
