# Peer review of "Expanding the Clinical Phenotype of 19q Interstitial Deletions: A New Case with 19q13.32-q13.33 Deletion and Short Review of the Literature"

_genes, 2022, doi:10.3390/genes13020212_

Round 1

Reviewer 1 Report

The deletion observed  by GCH array have to be confirmed by FISH.

Please add more information on the methods for the GCH analysis. 

Author Response

Thank you very much for reviewing our paper and for your comments! We will confirm this microdeletion using another method, unfortunately we haven't managed to find a laboratory which can perform this in just 5 days. We would like to kindly ask, if it's possible, for a month to do the confirmation and send the revised paper. We shall also complete the method for GCH annalysis.

Reviewer 2 Report

Line 145 should read 'which is missing.

Line 157-159 change to, An unrelated boy and girl had 19q13 with deletions overlapping SLO had 19q13 microdeletion phenotype. Namely,....................

Include materials and method,  array platform used, sensitivity and specificity of your validation for array detection. Are you using  the current 2020 ACMG aCGH  classification, FISH confirmation performed.

Author Response

Thank you very much for reviewing our paper and for your comments! We will confirm this microdeletion using another method, unfortunately we haven't managed to find a laboratory which can perform this in just 5 days. We would like to kindly ask, if it's possible, for a month to do the confirmation and send the revised paper. We shall also perform the solicited changes in lines 145 and 157-159.